# Germline Predisposition to Pediatric Cancer, from Next Generation Sequencing to Medical Care

**DOI:** 10.3390/cancers13215339

**Published:** 2021-10-24

**Authors:** Pablo Gargallo, Silvestre Oltra, Yania Yáñez, Antonio Juan-Ribelles, Inés Calabria, Vanessa Segura, Marián Lázaro, Julia Balaguer, Teresa Tormo, Sandra Dolz, José María Fernández, Carolina Fuentes, Bárbara Torres, Mara Andrés, María Tasso, Victoria Castel, Jaime Font de Mora, Adela Cañete

**Affiliations:** 1Pediatric Oncology Department, Hospital Universitario y Politécnico La Fe de Valencia, 46026 Valencia, Spain; yania_01@yahoo.com (Y.Y.); juan_antrib@gva.es (A.J.-R.); segura_van@gva.es (V.S.); balaguer_jul@gva.es (J.B.); maria_tormo@iislafe.es (T.T.); chemafer@mac.com (J.M.F.); fuentes_car@gva.es (C.F.); torres_bar@gva.es (B.T.); maradres@hotmail.com (M.A.); castel_vic@gva.es (V.C.); canyete_ade@gva.es (A.C.); 2Imegen–Health in Code Group, Department of Oncology, Paterna, 46980 Valencia, Spain; ines.calabria@imegen.es (I.C.); marian.lazaro@imegen.es (M.L.); 3Genetics Unit, Hospital Universitario y Politécnico La Fe de Valencia, 46026 Valencia, Spain; oltra_jua@gva.es; 4Genetics Department, Universidad de Valencia, 46010 Valencia, Spain; 5Laboratory of Cellular and Molecular Biology, Clinical and Translational Research in Cancer, Instituto de Investigación Sanitaria La Fe, 46026 Valencia, Spain; sandolzgi@gmail.com (S.D.); jaime.fontdemora@gmail.com (J.F.d.M.); 6Pediatric Oncology Department, Hospital General de Alicante, 03010 Alicante, Spain; tasso_mar@gva.es; 7Department of Pediatrics, Obstetrics and Gynecology, University of Valencia, 46010 Valencia, Spain

**Keywords:** genetic predisposition, genetic syndrome, pediatric oncology, germline, hereditary cancer, genetic counseling, working tool

## Abstract

**Simple Summary:**

The idea that motivated the design of the project is to offer a genetic germline analysis to all pediatric patients diagnosed in our pediatric oncology unit. The main objective is to determine the incidence of predisposing genetic variants when studying a cohort of pediatric cancer patients using an NGS gene panel. The custom panel employed is designed to detect variants in a large number of genes involved in pediatric cancer in order to be able to identify new genotype–phenotype relationships. The data obtained are valuable for estimating the incidence of predisposing genetic alterations, due to the large number of pediatric patients included in the study. Furthermore, the novel results collected in the main document, which suggest the involvement of new genes in the predisposition to different oncological diseases, are worthwhile.

**Abstract:**

Knowledge about genetic predisposition to pediatric cancer is constantly expanding. The categorization and clinical management of the best-known syndromes has been refined over the years. Meanwhile, new genes for pediatric cancer susceptibility are discovered every year. Our current work shares the results of genetically studying the germline of 170 pediatric patients diagnosed with cancer. Patients were prospectively recruited and studied using a custom panel, *OncoNano V2*. The well-categorized predisposing syndromes incidence was 9.4%. Likely pathogenic variants for predisposition to the patient’s tumor were identified in an additional 5.9% of cases. Additionally, a high number of pathogenic variants associated with recessive diseases was detected, which required family genetic counseling as well. The clinical utility of the Jongmans MC tool was evaluated, showing a high sensitivity for detecting the best-known predisposing syndromes. Our study confirms that the Jongmans MC tool is appropriate for a rapid assessment of patients; however, the updated version of Ripperger T criteria would be more accurate. Meaningfully, based on our findings, up to 9.4% of patients would present genetic alterations predisposing to cancer. Notably, up to 20% of all patients carry germline pathogenic or likely pathogenic variants in genes related to cancer and, thereby, they also require expert genetic counseling. The most important consideration is that the detection rate of genetic causality outside Jongmans MC et al. criteria was very low.

## 1. Introduction

Genetic predisposition plays an important role in cancer development. This fact is well-known in both the adult and pediatric patient population [1,2]. Environmental factors, involved in tumor onset during adult ages, are not so relevant in childhood [3]. However, the incidence and spectrum of mutations predisposing to cancer among children and adolescents are only partially understood [4,5,6]. Narod and colleagues claimed, in 1991, that 10% of children with cancer had a genetic predisposition [7]. Several genes related to predisposition to different childhood malignancies have been described since then (myeloid leukemia [8] and lymphoblastic leukemia [9], neuroblastoma [10], medulloblastoma [11,12,13], osteosarcoma [14] and soft tissue and bone sarcomas [15,16]). Meanwhile, knowledge on several disorders remains scarce, but current next generation sequencing technologies have expanded the frontiers of genetic predisposition research and, hence, the possibility of discovering new genotype–phenotype relationships [17].

Identifying cancer predisposition syndromes, defining them properly and establishing risk-adjusted surveillance programs are main goals of the scientific community [18,19]. Recent literature provided follow-up guidelines for several cancer predisposition syndromes with a broad consensus [20,21,22,23,24,25,26,27,28,29,30,31,32,33,34]. These recommendations open up the work of healthcare professionals in the case of detecting a genetic syndrome. Published guidelines are constantly being updated and are established as a useful framework for daily clinical practice.

The clinical feasibility of this genetic understanding is, therefore, clear and the advantages can be summarized in the following points: This knowledge enables a personalized medical and/or surgical treatment for several patients (e.g., Li–Fraumeni syndrome patients who have *TP53* mutations should not be exposed to ionizing radiation; surgical treatment should be conservative for hereditary retinoblastoma patients). It improves the selection of donors and the choice of the correct time for hematopoietic transplantation. This knowledge also allows to implement familial genetic counseling and transmit prognostic information to patients. It may accelerate the detection of associated non-tumor problems which may require early intervention (e.g., patients with a *WT1* mutation, who may have insidious renal dysfunction). Moreover, unraveling a genetic condition that explains the phenotype may help face the psychological burden of such a diagnosis in some patients/parents. Finally, it could provide a better understanding of tumor development in specific cases [35].

The growing knowledge on pediatric predisposition cancer syndromes underlines the great necessity to transfer into clinical practice the vast genetic knowledge generated by a genetic analysis. The present work aims to assess the incidence of genetic alterations in a prospective cohort of pediatric patients by a germline genetic analysis. We believe that at least 10% of our patients may suffer from a pediatric cancer predisposition syndrome. Therefore, we estimate that a relevant group of patients (at least 10%) could benefit from personalized follow-up recommendations or even personalized treatment. It is also expected to find at least 10% of families who could benefit from genetic counseling.

## 2. Materials and Methods

### 2.1. Patient Study Cohort

All potential candidate patients were diagnosed in or referred to our center from other hospitals between March 2018 and March 2020. Those who relapsed in our institution during this period were also considered for inclusion. Patient eligibility was assessed between days one and sixty since the first hospital admission. The following inclusion criteria were required for study entry:−Age between 0 and 18 years old.−Final pathology diagnosis established.−Germline origin blood sample availability.−Patient clinical stability.−Patient voluntary agreement to participate having understood the information related to the study.−No exclusion criteria fulfilled.

Furthermore, the exclusion criteria that conditioned the withdrawal of the study:−Rejection of the study by the patient and/or family.−Unfavorable previous psychological evaluation.−Diagnosis of a benign tumor without any known genetic basis for its development.

Patients who met the inclusion/exclusion criteria and agreed to enter the study were included. Patients who entered the project were clinically evaluated in a targeted way. A physical examination was performed and personal and family history were assessed, including a family tree. The information obtained was contrasted with Jongmans MC et al. criteria [36]. This tool allows the detection of patients who would benefit from personalized genetic counseling. In the event of any Jongmans MC et al. criteria being fulfilled and a genetic syndrome suspected, the patient was studied accordingly if a technique was available at the hospital (e.g., *RB1* by PCR and MLPA or *NF1/NF2* by a custom NGS panel). If no alterations were detected by these studies, the test was expanded by a custom NGS panel, *OncoNano V2*. Patients who did not fulfil those three conditions were studied by the *OncoNano V2* gene panel from the beginning. The workflow is shown in Figure 1.

The custom *OncoNano V2* panel was sequenced by the Institute of Genomic Medicine (Imegen-Health in Code Group). The technical report resulting from the genetic analyses was discussed by the Genetic Predisposition Committee of La Fe Hospital. A pediatric oncologist, a geneticist specialized in hereditary cancer and a molecular biologist were included in this committee. The final report prepared by the committee was delivered to the patient and family. Variants of uncertain significance were not communicated to the families. Pathogenic variants involved in the risk of cancer during childhood led to personalized recommendations for pediatric oncologists. Pathogenic or probably pathogenic variants with implications for cancer risk in adulthood guided family segregation studies and personalized follow-up of family members by the Genetic Counselling Unit. Likewise, pathogenic variants related to recessive diseases and their potential implications for the offspring were informed to the parents.

Patients and their families received pre- and post-testing genetic counseling. During the first visit, they were informed that the study consisted of sequencing a large NGS-based gene panel (390 genes), but that only a very low number of genes related to the pathology suffered by them or their children (score 1) was included in it. They were notified that exclusively the analysis of these genes, included as score 1, could obtain cause–effect conclusions for their individual cases. Moreover, they were advised that all the remaining data obtained from the analysis would be used for research purposes by the research team within the framework of the project. They were warned that some doubtful information may emerge from the study and that we could propose to continue studying different issues in the patient and/or family for research purposes. Nevertheless, in no case would this latest information allow us to obtain evidence for the specific patient. We were also committed not to harm or increase the number of patient and family medical visits when conducting these complementary tests. Thereupon, patients and/or parents signed the informed consent being aware of all this. Therefore, when identifying variants that were of interest from a research point of view, the families received the pertinent information during the post-testing visit. Accordingly, complementary studies (such as family segregation analysis) were carried out within this theoretical framework.

### 2.2. NGS Panel, Sequencing and Analysis Features

The *OncoNano V2* custom panel was developed in collaboration with Agilent and designed to detect mutations (point mutations, including single-nucleotide variants and small indels) and CNVs (deletions or duplications) in 390 genes related to pediatric cancer (Appendix A). The main established genes related to genetic predisposition to pediatric cancer were also covered. Genomic DNA (gDNA) from blood or other tissue was extracted using the commercial extraction kits RecoverAll™ and the QIAamp DNA Investigator Kit (QIAGEN, Hilden, Germany). Concentration was measured by fluorometric quantification using a Qubit fluorometer with the Qubit dsDNA BR Assay kit (Thermo Fisher Scientific, Waltham, MA, USA) and Qubit dsDNA HS Assay kit (Invitrogen, Waltham, MA, USA). DNA Integrity Number (DIN) was determined using the DNA ScreenTape assay (Agilent Technologies, Santa Clara, CA, USA). The cut-off DIN value was 3. Library preparation followed the manufacturer’s recommendations. Libraries were then loaded onto the NextSeq 550 system (Illumina, San Diego, CA, USA) for massive library sequencing in “Stand-alone” mode with 2 × 150 paired-end reads following the manufacturer’s instructions. For bioinformatics analysis, the alignment to the reference sequence Genome Reference Consortium Human Build 37 (GRCh37), annotation and variant calling followed a custom pipeline through the DataGenomics platform by Imegen. For the CNV analysis, in-house scripts by Imegen were used to obtain a fractional coverage based on a correlation between the number of normalized reads of a region in respect to the number of DNA copies for that region. A minimum inter-sample variability was guaranteed by homogenizing experimental conditions between different samples and genomic regions. CNV calls were classified by DataGenomics based on their credibility, using a scoring algorithm that took into account parameters such as log2 ratio, event size, proximity and type of contiguous events. CNV plots provided by the platform were manually reviewed to discard possible artifacts and validated by digital PCR or MLPA.

The panel genes were classified into 3 scores, depending on their involvement at the hereditary cancer level in order to facilitate further manual analysis. Genes involved in predisposition to the patient’s tumor were studied as score 1. Genes involved in predisposition to other tumors as score 2 and other genes related to pediatric cancer at the somatic level and included in the panel were included as score 3. The analysis was performed with the DataGenomics software. Filters were applied to remove from the analysis variants with an MAF (minor allele frequencies) > 0.02 and variants in non-coding regions (flanking splicing sites up to ±10 nucleotides were excluded from filters). Changes described as polymorphic according to gnomAD browser data were also removed from the analysis. The study of the variants was carried out with the help of the VarSome, COSMIC, professional HGMD and Pcan.stjude.org websites, as well as those available for specific genes. Information obtained from in silico predictions was also considered.

The variants were classified as benign, likely benign, VUS (variant of uncertain significance), likely pathogenic (LP) and pathogenic (P) following ACMG recommendations [37]. In addition, some VUSs were considered to be potentially involved in genetic predisposition to the disease. However, for many of them, evidence was scarce in this clinical context. Despite a comprehensive in silico analysis, a review of the available literature and a discussion of the variants in expert committee, no strong conclusions could be drawn. For these variants of uncertain significance, potentially involved in predisposition to the patient’s cancer but without enough evidence to be considered probably pathogenic, an internal nomenclature was established. Variants of potential pathogenic significance (VOPPS) was the term used for these variants.

## 3. Results

### 3.1. Patient Cohort and Genetic Variants Identified

Overall, 223 patients were assessed for inclusion during the specified period. Finally, 170 patients fully met the inclusion criteria and agreed to participate in the study. The parents signed the informed consent in all cases, but the patients were also informed according to their ages and patients older than 12 years signed specific documents.

The male–female distribution was 60–40% and the mean age was 7.2 years (0–18). The most common diagnosis was leukemia (45 cases; 26.5%), followed by CNS tumors (26 cases; 15.3%), lymphomas (20; 11.8%), neuroblastoma and peripheral nervous system tumors (19; 11.2%), bone tumors (14; 8.2%), soft-part sarcomas (12; 7.1%), renal tumors (9; 5.3%), retinoblastoma (8; 4.7%), liver tumors (4; 2.3%), germ-cell tumors (3; 1.8%), melanoma and other skin tumors (1; 0.5%) and other tumor types (9; 5.3%) (Figure 2).

These percentages were compared to those collected in the Spanish Registry of Pediatric Tumors (RETI) [38]. Statistically significant differences were not found when comparing the incidence rates for these tumor types between the RETI series (age group 0–19 years; 1980–2017) and our cohort. Following the workflow established, a total of 153 patients was studied with *OncoNano V2*, and the remaining 17 cases exclusively by conventional techniques or other NGS panels (Appendix A).

A pathogenic variant predisposing to the patient’s tumor was detected in 16 cases (16/170; 9.4%). Regarding the genes involved in predisposition, the most frequently altered was the *RB1* gene (6/16; 37.5%), followed by *NF1* (3/16; 18.8%); other mutated genes were *DICER1, NF2, SUFU, TP53, XPC* and *SOS1*. Moreover, a patient diagnosed with trisomy 21 was included in the cohort. In addition, likely pathogenic mutations that could be involved in predisposition to the patient’s tumor were identified in ten other cases (10/170; 5.9%). These 26 pathogenic and likely pathogenic variants detected are summarized in Table 1 and Figure 3.

Other pathogenic/likely pathogenic variants were not considered to be involved in predisposition to patient tumors because they were related to recessive diseases, but without any evidence to associate them with the patient’s cancer. However, their involvement cannot be ruled out in certain cases: *ERCC3* (patient 159; Ewing sarcoma), *XPC* (patient 151; ependymoma), *FANCM* (patient 149; neuroblastoma), *PIK3CG* (patient 111; lymphoma), *RECQL4* (patient 89; leukemia), *NBN* (patient 78; atypical teratoid rhabdoid tumor), *FANCL* (patient 42; rhabdomyosarcoma) and *CEP57* (patient 18; astrocytoma) (details on the variants can be found in Table 2 and Appendix A). Besides that, some VUSs might be attributed to a potential pathogenicity. Hence, they might be involved in predisposition to the tumor suffered by the patients, but the lack of evidence leads to classifying them as VUSs according to the ACMG criteria. These variants were classified as VOPPS. Variants of these characteristics were detected in the genes *ING4* (patient 153; carcinoid tumor), *NF1* (patient 144; neuroblastoma), *FANCD2* (patient 143; lymphoma), *IGF1R* (patient 139; Wilms tumor), *ALK* (patient 119; leukemia), *FAT1* (patient 81; HGG), *CHEK2* (patient 78; teratoid/rhabdoid tumor), *RET* (patient 72; leukemia) and *SH2B3* (patient 48; leukemia) (more in Table 2; Appendix A). Variants of uncertain significance or likely benign not previously reported in databases or with a higher incidence than expected in cancer patients were also collected in Appendix A.

Overall, and considering all P/LP variants identified, related or not to genetic predisposition to patient’s tumor, 35 out of 170 patients/families (20.6%) carried at least one of these variants. Families received this information and adequate genetic counseling.

### 3.2. Jongmans MC et al., 2016, Tool Evaluation

A total of 50 patients (29%) met the indication for referral to a clinical geneticist according to the Jongmans MC et al. criteria during the targeted assessment carried out after inclusion. Among them, pathogenic predisposing mutations were detected in 15 cases (15/50; 30%). It can be seen from this that 94% of the total of pathogenic variants predisposing to pediatric cancer detected in the study (15/16) was found in patients who met the Jongmans MC et al. criteria. In addition, five out of ten variants (50%), classified as likely pathogenic, were detected among patients who met the Jongmans MC et al. criteria. Therefore, pathogenic and likely pathogenic mutations were identified in 40% of the patients chosen by the tool (20/50). Considering predisposition variants as only the 16 pathogenic mutations, the Jongmans MC et al. tool was found to have a sensitivity of 94% and a specificity of 77% in our cohort. Taking into consideration both pathogenic and likely pathogenic variants probably involved in predisposition, the sensitivity would be 77% and specificity 79%. For the 120 patients who did not meet the indication for referral to a clinical geneticist based on the Jongmans MC et al. criteria, pathogenic predisposition mutations were detected only in one case (0.8%). Out of the 120 patients, five carried likely pathogenic variants according to the ACMG criteria (4.2%).

The phenotype–genotype correlations of those patients carrying likely pathogenic variants are described below:

### 3.3. CTCF Variant c.1337-T>A and Wilms Predisposition

Patient number 105 corresponds to a 2-year-old child diagnosed with bilateral Wilms tumor. The phenotype was intellectual development at the limit of normality, bilateral cryptorchidism, patent foramen ovale, minor facial dysmorphism, such as a prominent forehead, leafy and arched eyebrows, a long filtrum and thin upper lip. Therefore, the evaluation using the Jongmans MC et al. tool was positive; however, it did not suggest any diagnosis. The NGS study identified the likely pathogenic *CTCF* variant c.1337-T>A (p.I446K) (NM_006565.4), with an allelic frequency of 50%. This variant was confirmed in homozygosity both in tumor DNA and RNA. The family segregation study confirmed that the variant occurred de novo in the patient. The detection of this variant in the clinical context of the patient, having adequately ruled out other entities predisposing to Wilms tumor, led us to the diagnosis of mental retardation, autosomal dominant 21 [39]. After a multidisciplinary assessment, we considered that this variant might predispose to Wilms tumor in the context of MRD21; this tumor has not been reported in other MRD21 patients to date.

### 3.4. BRCA1 c.68_69del Variant and Neuroblastoma Susceptibility

Patient number 59 was diagnosed with poorly differentiated mediastinal neuroblastoma at the age of 6 months (*NMYC* not amplified; without segmental chromosomal alterations in SNP Array). Parents were consanguineous, but data suggestive of a familial predisposition syndrome were not detected. The evaluation using the Jongmans MC tool was positive (consanguinity). An NGS study was carried out since there was no suspicion of a specific entity. The *BRCA1* variant c.68_69del e.E23Vfs*17 (NM_007294.3) was detected in heterozygosity. The relationship between *BRCA1* mutations and predisposition to neuroblastoma is based on casual findings in specific cases, such as our patient (10). The *BRCA1*–Neuroblastoma risk ratio is still under study; therefore, the implications of this variant in tumor development are currently undetermined. The parents refused the family segregation study and no additional family information was provided.

### 3.5. CHEK2 c.497A>G Variant and B-Cell ALL Risk

Patient number 118 suffered from B-cell acute lymphoblastic leukemia when he was one year old, without other remarkable personal clinical data. Her mother had breast cancer at age 41 and a non-informative *BRCA1* and *BRCA2* study result. Colon cancer in her grandfather on the mother’s side at age 73 stood out in the family history, as well as Hodgkin’s lymphoma at age 45 in one of the mother’s three siblings. There are no cases of cancer reported on the father’s side. The NGS study identified heterozygous *CHEK2* c.497A>G (p.N166S) NM_007194.3. A segregation study confirmed the maternal origin of the variant and the remaining members of the family are under study. Based on the evidence available for *CHEK2* mutations in breast cancer, this variant might be involved in the mother’s breast cancer [40]. However, evidence supporting the relationship between *CHEK2* variants and the risk of leukemia is still limited [41].

### 3.6. CDKN2A Deletion and Leukemia

Patient number 110 was diagnosed with common B-cell ALL at the age of 3 years. Her mother was diagnosed with metastatic melanoma and died of the disease at a young age. Following the established workflow, the *OncoNano V2* panel was sequenced. A mono-allelic *CDKN2A* deletion was detected and it was confirmed by MLPA. There was a high probability that the *CDKN2A* deletion was inherited from the mother, but it could not be confirmed. While the relationship between melanoma and *CDKN2A* is well known, information on the involvement of the *CDKN2A* gene in leukemia predisposition is scarce. However, a possible association of some *CDKN2A* polymorphisms (rs3731249 and rs3731217) with ALL risk in pediatric age has been proposed [42,43,44]. In this context, we concluded that the detected deletion might have facilitated the tumor development in the patient, although currently available evidence is insufficient.

### 3.7. JAK3 Mutations and Familial Leukemia

Patient number 116 was a 5-year-old girl with a diagnosis of common B-ALL (acute lymphoblastic leukemia). Her mother also had ALL at the age of 5 years. The mother survived and was now 36 years old. The remaining information available on the maternal side was not contributory. Given that the patient met the Jongmans MC et al. criteria, the patient’s sample was sequenced. Two CIS heterozygous and likely pathogenic variants were identified in the *JAK3* gene. The detected variants were *JAK3* c.1465C>T (p.Q489*) and *JAK3* c.1442-2A>G. A family segregation study of both variants was completed. The mother carried both variants, while the father did not. The *JAK3* c.1442-2A>G variant, located closer to the N-terminal end than the other variant, was thought to be a null variant leading to the loss of protein function. Loss-of-function mutations in homozygosity or compound heterozygosity are associated with severe combined immunodeficiency, whose inheritance is autosomal recessive. However, heterozygous loss-of-function mutations have not been associated with leukemia predisposition to date. Given the peculiarity of the family history, we considered the variant(s) to be likely pathogenic. Whether the variant(s) was involved in predisposition to leukemia suffered by the mother and daughter is completely unknown.

Other data of interest related to these and other specific cases carrying LP variants are shown in Appendix A and Figure 4. Some of the complementary studies carried out in specific patients or families in response to the detection of some variants are contained in Appendix A.

## 4. Discussion

This study presented the results of an *OncoNano V2* NGS panel sequencing of germline samples from a large cohort of pediatric oncology patients. On the basis of our results, it can be concluded that up to 9.4% patients had a genetic predisposition syndrome which explained the cancer they suffered. Meanwhile, considering that an additional 5.9% of the patients carried likely pathogenic variants, a few of which might be involved in susceptibility to the disease, this figure might be higher. The results obtained were consistent with previously published data [4,5,6]. Recent results from the MSK-IMPACT cohort also pointed the same way [46]. New predisposition genes have been described in the last two years [12] and these genes were not included in the panel; therefore, the figure presented might be considered conservative.

Due to the high number of patients to be assessed from a germline point of view, selection tools to enable a concise assessment may improve the decision making in this field. We evaluated the usefulness of the Jongmans MC et al. tool for this purpose in our cohort. The tool showed a high sensitivity for the detection of patients with currently well-categorized predisposition syndromes in our cohort (94%). A Li–Fraumeni patient was the only case of a well-established and undetected genetic syndrome by the tool. He was a 5-year-old patient diagnosed with anaplastic embryonal rhabdomyosarcoma. No other data of interest were found in the medical records. The presence of a family history of cancer was ruled out. A variant considered pathogenic was detected in the *TP53* gene and the diagnosis of Li–Fraumeni syndrome was determined. Family studies could not be expanded because of the early death of the patient and loss of contact with parents. The association between soft-tissue sarcoma and Li–Fraumeni syndrome is high [47]. The benefit of studying *TP53* at least in patients with anaplastic rhabdomyosarcoma is reinforced by current evidence [48]. Ripperger T et al. modified the Jongmans MC criteria [18] and their updated tool would have detected this case; this tool had achieved a sensitivity of 100% in the detection of pathogenic variants within the analyzed cohort in this study. The Ripperger T et al. revision also added rare entities specific to cancer predisposing syndromes in order to improve the sensitivity of the selection tool (e.g., Botryoid rhabdomyosarcoma of the urogenital tract). The inclusion of patients with acute myeloid leukemia, based on the 2016 WHO recommendations, was also considered [49]. Currently, this would be the most appropriate tool for patient selection in order to recommend a genetic study.

Our results highlighted the challenge of interpreting genetic variants in the context of predisposition to pediatric cancer. The cases carrying LP variants and above presented were only an example of frequent difficulties found throughout the series. Variants in genes *CHEK2, MRE11, PALB2 and ATM* reported for patients 15, 36, 39, 89 and 120 presented similar challenges. These clinical cases required constant re-evaluation based on the evidence available at any given time. The same was true for rare VOUS in the general population, especially those to which we attributed potential pathogenicity and designated VOPPS (variants located in *ING4, IGF1R, NF1, FANCD2, ALK, CHEK2, FAT1* and *SH2B3* genes). This subsequent work should be considered from the beginning in order to properly quantify the resources that will be required in the long term. Despite the limitations found for variant interpretation, this work has allowed to detect genetic variants that might be related to new genotype–phenotype associations for different pathologies. These data could be investigated in larger patient cohorts by international collaborative groups.

This kind of clinical approximation, with so many personal and family implications, demands a comprehensive assessment of the advantages and disadvantages detected. The NGS technology used allowed us to reliably test for SNVs and CNVs in 390 gene regions in a single test. Despite the lower cost and accessibility of this technology at present, a cost/effectiveness assessment should be carried out. Based on the results obtained, the detection of the main pediatric cancer predisposition syndromes could be possible through a considerably smaller and less expensive gene panel than *OncoNano V2.* A pre-test approach based on a tool such as that of Jongmans MC et al. or Ripperger T et al. could achieve an adequate selection of most of the patients that should be studied. In fact, one of the main conclusions raised from this work was that, outside of Ripperger T et al. criteria and, therefore, out of the syndromes included in their review, no genetic alteration with evidence of being responsible for the disease suffered by the patient was identified. Accordingly, sequencing broad panels such as ours or WES would make sense only and exclusively in the field of research or when facing extremely particular clinical cases. Therefore, for daily clinical practice and in order to detect cancer predisposition syndromes, the analysis of genes not related to the entities collected by Ripperger T et al. gave rise to more doubts than certainties. In consequence, their testing would not be recommended outside the research field.

On the other hand, pre- and post-psychological test assessments play a key role in proper long-term management. This has been proved as something basic in different areas of genetics [50] and is especially important in this field, with so many consequences in the personal and family sphere. Moreover, the turnaround time required for suitable clinical implementation is a huge challenge in this context. Workflow based on a pre-test clinical and psychological evaluation, patient/family information in a context of high emotional stress, germline sample collection, sequencing and analysis, committee discussion and reporting require a constant evaluation of deadlines. Undoubtedly, if the turnaround time is too long, some of the potential clinical benefit may be lost. However, the optimal time mainly depends on the specific case of each tumor type, or even of each patient. This work also highlighted the importance of an expert committee, and not only in the field of research, since the challenges derived from studying relatively well-known genes and syndromes remain remarkable. This multidisciplinary approach has already been shown to be very useful in other fields of personalized medicine [51], but it slowly emerges as a key component in this research area. Nonetheless, expert knowledge of each of the analyzed genes is an arduous task for any human group. Therefore, consultation with other national and international experts is presented as a useful tool in this field. The implementation of networks focusing on pediatric cancer predisposition syndromes will be essential in the following years.

## 5. Conclusions

In summary, it should be noted that, in nearly 20% of the patients, genetic data were identified that could have personal and family implications. In a few of them (9.4%), a genetic syndrome was diagnosed; thereby, the information was clinically useful for the patient. However, uncertainty was transferred to families in several cases when analyzing genes previously unrelated to pediatric cancer predisposition syndromes. In fact, outside of the genes and syndromes included in the Ripperger T et al. criteria, not a single cancer predisposition syndrome was identified in this study. For this purpose, it seems preferable in clinical practice to sequence a highly selected gene panel rather than a large one or the whole exome after evaluating and choosing the patients with a selection tool such as that of Jongmans MC et al. or Ripperger T et al.

## Figures and Tables

**Figure 1 cancers-13-05339-f001:**
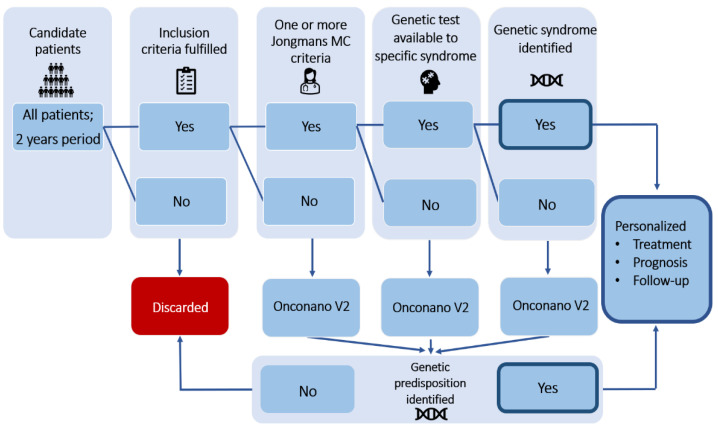
Workflow established for the study.

**Figure 2 cancers-13-05339-f002:**
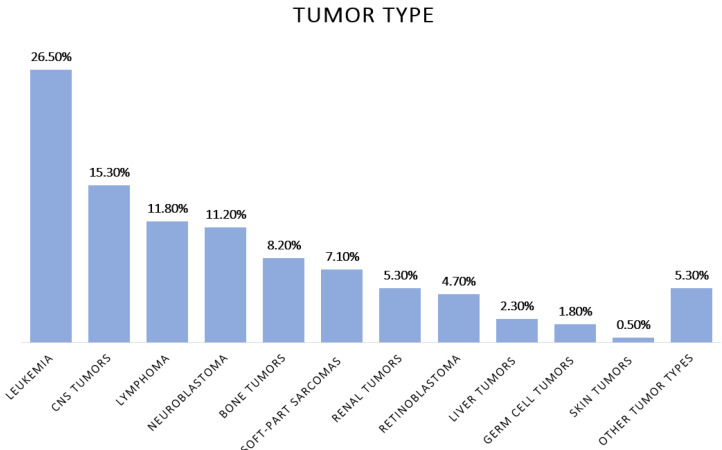
Distribution of patients (%) by tumor type.

**Figure 3 cancers-13-05339-f003:**
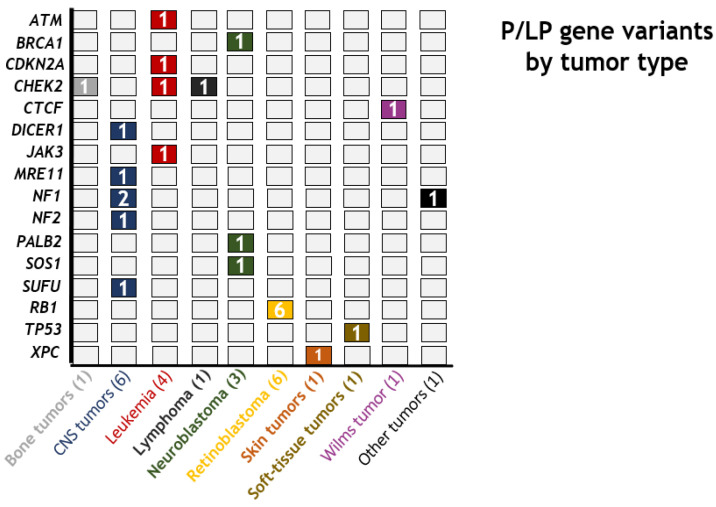
Genes and tumor types whereby pathogenic or likely pathogenic variants were identified.

**Figure 4 cancers-13-05339-f004:**
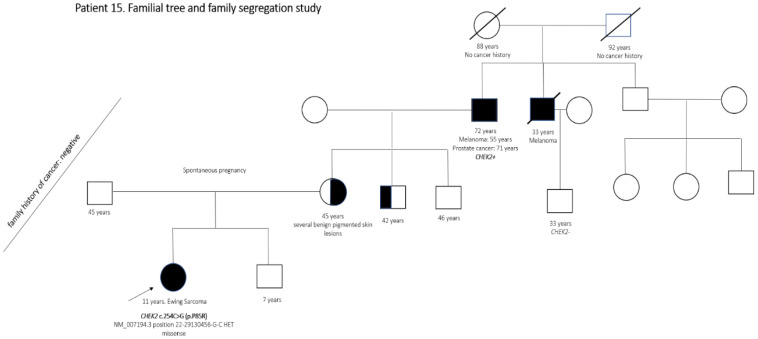
Family tree of patient number 15. Despite not fulfilling the Jongmans MC criteria nor the revised criteria by Ripperger, the patient’s family history of cancer was still suggestive for genetic cancer predisposition and the genetic counseling was advised based on this information. The adolescent was diagnosed with extraosseous Ewing sarcoma. A *CHEK2* variant was detected by the NGS *OncoNano V2* panel. The variant was described in the general population (gnomAD reports three total heterozygotes). However, it was described six times in ClinVar and three times in cancer patients (uncertain clinical significance (ID 233261)). It was not found in other databases. In addition, it is a variant studied functionally on one occasion. It was reported in the literature in a patient with hereditary breast cancer, with functional in vitro study that demonstrated a 50% reduction in kinase activity [45], although the location of the variant was outside of a functional domain. The variant was found to be of maternal origin and family history of melanoma was identified in the grandfather and great-uncle in this branch of the family. In addition, the grandfather had had a second tumor at an older age. Based on the ACGM criteria and family information, the variant was classified as likely pathogenic.

**Table 1 cancers-13-05339-t001:** Pathogenic or likely pathogenic variants considered to be involved (pathogenic) or maybe involved (likely pathogenic) in patient’s disease.

Patient Number	Diagnosis	Gene Variant/Genomic Alteration	Categorization
10	Retinoblastoma (unilateral)	13q12.13-q21.2 deletion	Pathogenic
11	Retinoblastoma (unilateral)	*RB1* c.844G>T (p.E282*)	Pathogenic
12	Pilocytic astrocytoma	*NF1* C.910C>T (p.R304*)	Pathogenic
15	Ewing sarcoma	*CHEK2* c.254C>G (p.P85R)	Likely pathogenic
31	Neuroblastoma	*SOS1* c.1300G>A (p.G434R)	Pathogenic
36	Pilocytic astrocytoma	*MRE11* c.659+1G>A	Likely pathogenic
39	Neuroblastoma	*PALB2* c.2747A>T (p.E916V)	Likely pathogenic
44	B-ALL	Trisomy 21	Pathogenic
51	Retinoblastoma (bilateral)	*RB1* c.2104 C>T (p.Q702*)	Pathogenic
59	Neuroblastoma	*BRCA1* c.68_69del (p.E23Vfs*17)	Likely pathogenic
64	Retinoblastoma (unilateral)	13q12q21 deletion	Pathogenic
65	Plexiform neurofibroma	*NF1* c.4084C>T (p.R1362*)	Pathogenic
66	Retinoblastoma (bilateral)	*RB1* c.224G>A (p.W75*)	Pathogenic
89	B-ALL	ATM c.1402_1403del (p.K468Efs*18)	Likely pathogenic
103	Cutaneous angiosarcoma	*XPC* c.1643_1644delTG (p.V548Afs*25) (homozygous)	Pathogenic
105	Wilms tumor	*CTCF* c.353T>A (p.I118K)	Likely pathogenic
108	Embryonal rhabdomyosarcoma	*TP53* c.559G>A (p.G187S)	Pathogenic
110	B-ALL	*CDKN2A* deletion	Likely pathogenic
113	Medulloblastoma SHH	*SUFU* c.71dup (p.A25Gfs*23)	Pathogenic
116	B-ALL	*JAK3* c.1465C>T (p.Q489*) and *JAK3* c.1442-2A>G	Likely pathogenic
118	B-ALL	*CHEK2* c.497A>G (p.N166S)	Likely pathogenic
120	Burkitt lymphoma	*CHEK2* c.470T>C (p.I157T)	Likely pathogenic
127	Schwannoma CNS (NF1 phenotype) **	*NF1* c.2251+1 G>A	Pathogenic
150	Vestibular schwannoma (bilateral)	*NF2* c.115-2A>G	Pathogenic
156	Pineoblastoma	*DICER1* c.2026C>T (p.R676*)	Pathogenic
169	Retinoblastoma (bilateral)	*RB1* c.2548_2552delCAGA-T (Q850Gfs*3)	Pathogenic

** Schwannoma is not an NF1 feature, but the patient fulfilled an NF1 diagnosis according to the NIH criteria with >6 café-au-lait spots (CAL), axillary freckling and a proven neurofibroma.

**Table 2 cancers-13-05339-t002:** Pathogenic or likely pathogenic variants considered as not involved in the tumor etiology by the Pediatric Cancer Predisposition Committee. Variants of uncertain significance to which potential pathogenicity was attributed by the committee (variants of potential pathogenic significance—VOPPS).

**P/LP Variants Not Predisposing to Patient’s Tumor**		
**Patient Number**	**Diagnosis**	**Gene Variant**
18	Pilocytic astrocytoma	*CEP57* c.241C>T (p.R81*)
42	Alveolar rhabdomyosarcoma	*FANCL* c.40del (p. L14Cfs*27)
42	Alveolar rhabdomyosarcoma	*XPC* c.1643_1644del (p. V548Afs*25)
78	Atypical teratoid rhabdoid tumor	*NBN* c.1648_1651del (p. K550Gfs*8)
89	B-ALL	*RECQL4* c.2336_2357del (p.D779Cfs*57)
111	Lymphoblastic lymphoma	*PIK3CG* c.2340dup (p. E781Rfs*4)
149	Neuroblastoma	*FANCM* c.2161-1G>A
151	Ependymoma	*XPC* c.1643_1644del (p. V548Afs*25)
159	Ewing sarcoma	*ERCC3* c.583C>T (p. R195T*)
**VOPPS Variants**		
**Patient Number**	**Diagnosis**	**Gene Variant**
48	B-ALL	*SH2B3* c.622G>C (p.E208Q)
72	B-ALL	*RET* c.2331C>A (p.N777K)
78	Atypical teratoid rhabdoid tumor	*CHEK2* c.342G>T (p.W114C)
81	High grade glioma	*FAT1* c.10990del (p.Q3664Sfs*10)
119	B-ALL	*ALK* c.3467G>A (p.C1156Y)
139	Wilms tumor	*IGF1R* c.3367A>G (p.M1123V)
143	Lymphoblastic lymphoma	*FANCD2* c.2204G>A (p.R735Q)
144	Neuroblastoma	*NF1* c.2998C>A (p.R1000S)
153	Carcinoid tumor	*ING4* c.109+1G>C

## Data Availability

All clinical and genetic information derived from the study was included in the paper itself, as well as in the Appendix A.

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
