# Peer review of "Germline Predisposition to Pediatric Cancer, from Next Generation Sequencing to Medical Care"

_cancers, 2021, doi:10.3390/cancers13215339_

Round 1
Reviewer 1 Report
The authors have now made appropriate changes and the article is much improved. I still feel a footnote to the table is required
Schwannoma diagnosis does NOT support an NF1 diagnosis. How did the patient
meet NF1 criteria
The patient was diagnosed of NF1 syndrome diagnosis according to NIH criteria:
-More than six CAL macules (the greatest diameter of which is more than 5mm (the
patient is 6 years old)
-Axillary freckling
Moreover, she has one neurofibroma in her left side back
Please put a footnote to table to state that although schwannoma is not a feature of NF1 the patient met NIH criteria with >6CAL, freckling and a proven neurofibroma. Ideally the schwannoma diagnosis should be reviewed as schwannoma features usually in an NF1 tumor are now more correctly diagnosed as a hybrid tumor
Author Response
thank you very much for your work on this manuscript. We have added this footnote.

Reviewer 2 Report
The author's have again improved their manuscript which is in general suitable for publication. However, some minor changes are suggested:
Line 179 - Please Introduce the abbreviated term "MAF".
Figure 4: It should be started that, although not fulfilling the criteria by Jongmans nor the revised criteria by Ripperger, the patient's family history of cancer is still suggestive for genetic cancer predisposition and genetic counseling might be advised just by that information.
Lines 378+: It should be stated that the Ripperger tool had a sensitivity of 100% in the detection on proven pathogenic variants its the analyzed cohort.
(Just for information: This year, there has been an update of the Ripperger tool which is recommender to be used in Germany by the GPOH.)
Author Response
The author's have again improved their manuscript which is in general suitable for publication. However, some minor changes are suggested: thank you very much for your work on this manuscript.
Line 179 - Please Introduce the abbreviated term "MAF".: introduced, thank you.
Figure 4: It should be started that, although not fulfilling the criteria by Jongmans nor the revised criteria by Ripperger, the patient's family history of cancer is still suggestive for genetic cancer predisposition and genetic counseling might be advised just by that information.: Added, thank you
Lines 378+: It should be stated that the Ripperger tool had a sensitivity of 100% in the detection on proven pathogenic variants its the analyzed cohort.: included, thank you
(Just for information: This year, there has been an update of the Ripperger tool which is recommender to be used in Germany by the GPOH.): thank you very much for this information.

This manuscript is a resubmission of an earlier submission. The following is a list of the peer review reports and author responses from that submission.
Round 1
Reviewer 1 Report
The authors present the results of germline NGS testing of an unselected series of children with malignant tumors including benign CNS tumors. The authors try to make to much out of probably chance findings of a common BRCA1 and CHEK2 variant as well as a number of other variants that are not definitively associated with the tumor being assessed. The main finding is actually that Jongmans’ criteria are extremely effective in identifying children who should receive a panel. With 30% detection rate of definitely associated genetic predisposition compared to only 1/120 of the remainder. Given the complexity of genetic counselling and potential for overinterpretation that the authors themselves have fallen into the main conclusion should be that testing outside Jongmans’ criteria should only be done on a research basis to determine potential links. The authors should provide more information on those with definite gene associations as to whether they met clinical criteria such as bilateral retinoblastoma, NF1 and NF2. The abstract discussion and conclusions all need tempering to reduce the slight overenthusiasm. Nevertheless, with a more careful assessment of the data and a change in emphasis this may still be a useful contribution. For instance the findings relating to the CTCF Variant c.1337-T > A and Wilms Predisposition is quite compelling and there are previous studies to suggest the potential importance of CCTF in Wilms. Overall the tenor of the entire paper and conclusions must be rewritten as is the main focus on stating how ‘useful’ the panels are.
Specific comments
- ‘However, for several disorders, knowledge remains scarce which is the case of many soft tissue and bone sarcomas [15, 16].’ -It is well known that germline TP53 predisposes to both
- ‘Genes involved in predisposition to the patient's tumor were studied as score 1. Genes involved in predisposition to other tumors, as score 2 and other genes in the panel were included in score 3’ Why were score 3 included in the panel if they were not known to be associated with childhood cancer?
- ‘. Finally, 170 patients fully met the inclusion criteria and agreed to participate in the study.’ -I presume it was the parents that agreed?
- ‘Other mutated genes were DICER1, NF2, SUFU, TP53, XPC, SOS1, along with trisomy 21.’ Trisomy 21 is not a ‘mutated’ gene. Presumably this was known clinically? Was the panel actually used for this and did it pick this up?
- ‘Regarding the genes involved in predisposition, the most frequently altered was the RB1 gene (6/16; 37.5%) followed by NF1 (3/16; 18.8%)’ -How many were already diagnosed clinically eg bilateral RB and NF1.
- Table S1 please state if retinoblastoma bilateral or multifocal, Please confirm if really a ‘schwannoma’ with NF1. Please confirm if an ‘isolated’ VS in NF2. Schwannomas are not a feature of NF1 these are nearly always neurifibroma-was this biopsied?
- I think it is a serious push to have CHEK2 and ATM heterozygotes especially CHEK2 missense as being considered involved in table S1
- ‘Pathogenic or likely pathogenic variants considered as no(t) involved in the tumor etiology by the Pediatric Cancer Predisposition Committee. 2B.’ -‘not’
- ‘Overall and considering all P/LP variants identified, related or not to genetic predisposition to patient´s tumor, 40 out of 170 patients/families (23.5%) carried at least one of 232 these (mutations)’ Variants NOT ‘mutations’
- ‘The relationship between BRCA1 mutations and predisposition to neuroblastoma is based on casual findings in specific cases, such as our patient (10). Although the BRCA1-Neuroblastoma risk ratio is still under study, we consider this variant to be likely involved in neuroblastoma predisposition. The parents refused the family segregation study and no additional family information was provided. ‘ I seriously disagree with this ‘likely involved’ conclusion. This is a common founder variant carried by 1% of Jewish people. If the parents are Jewish this is a likely chance finding
- ‘ The NGS study identified heterozygous CHEK2 c.497A>G (p.N166S) NM_007194.3. Segregation study confirmed the maternal origin of the variant and the remaining members of the family are under study. The evidence available in breast cancer led us to suggest the involvement of the variant in the mother's breast cancer [40]. However, evidence supporting the relationship between CHEK2 variants and the risk of leukemia is still limited [41]. Even so, we considered this variant to be likely involved in susceptibility to leukemia in the pediatric age.’ This consultant in Medical Genetics again does not agree with this. Even the breast cancer link for missense variant is associated with only a 1.6-1.7 relative risk meaning most breast cancers will be coincidental. This is not good clinical governance to make these decisions that are likely misleading
- ‘On the basis of our results, it can be concluded that at least 9.4% patients have a genetic predisposition syndrome which explains the cancer they suffer’ A separate table of these 16 with references to the evidence base for them being definitively associated is required
- The abstract discussion and conclusions all need tempering to reduce the slight overenthusiasm. In particular the main highlight is actually the low useful detection rate of definitive genetic causality outside Jongmans. The lesson of this is that potentially more harm may be done by speculation about causality where there is little evidence especially outside Jongmans criteria cases.
The authors present the results of germline NGS testing of an unselected series of children with malignant tumors including benign CNS tumors. The authors try to make to much out of probably chance findings of a common BRCA1 and CHEK2 variant as well as a number of other variants that are not definitively associated with the tumor being assessed. The main finding is actually that Jongmans’ criteria are extremely effective in identifying children who should receive a panel. With 30% detection rate of definitely associated genetic predisposition compared to only 1/120 of the remainder. Given the complexity of genetic counselling and potential for overinterpretation that the authors themselves have fallen into the main conclusion should be that testing outside Jongmans’ criteria should only be done on a research basis to determine potential links. The authors should provide more information on those with definite gene associations as to whether they met clinical criteria such as bilateral retinoblastoma, NF1 and NF2. The abstract discussion and conclusions all need tempering to reduce the slight overenthusiasm. Nevertheless, with a more careful assessment of the data and a change in emphasis this may still be a useful contribution. For instance the findings relating to the CTCF Variant c.1337-T > A and Wilms Predisposition is quite compelling and there are previous studies to suggest the potential importance of CCTF in Wilms. Overall the tenor of the entire paper and conclusions must be rewritten as is the main focus on stating how ‘useful’ the panels are.
Specific comments
- ‘However, for several disorders, knowledge remains scarce which is the case of many soft tissue and bone sarcomas [15, 16].’ -It is well known that germline TP53 predisposes to both
- ‘Genes involved in predisposition to the patient's tumor were studied as score 1. Genes involved in predisposition to other tumors, as score 2 and other genes in the panel were included in score 3’ Why were score 3 included in the panel if they were not known to be associated with childhood cancer?
- ‘. Finally, 170 patients fully met the inclusion criteria and agreed to participate in the study.’ -I presume it was the parents that agreed?
- ‘Other mutated genes were DICER1, NF2, SUFU, TP53, XPC, SOS1, along with trisomy 21.’ Trisomy 21 is not a ‘mutated’ gene. Presumably this was known clinically? Was the panel actually used for this and did it pick this up?
- ‘Regarding the genes involved in predisposition, the most frequently altered was the RB1 gene (6/16; 37.5%) followed by NF1 (3/16; 18.8%)’ -How many were already diagnosed clinically eg bilateral RB and NF1.
- Table S1 please state if retinoblastoma bilateral or multifocal, Please confirm if really a ‘schwannoma’ with NF1. Please confirm if an ‘isolated’ VS in NF2. Schwannomas are not a feature of NF1 these are nearly always neurifibroma-was this biopsied?
- I think it is a serious push to have CHEK2 and ATM heterozygotes especially CHEK2 missense as being considered involved in table S1
- ‘Pathogenic or likely pathogenic variants considered as no(t) involved in the tumor etiology by the Pediatric Cancer Predisposition Committee. 2B.’ -‘not’
- ‘Overall and considering all P/LP variants identified, related or not to genetic predisposition to patient´s tumor, 40 out of 170 patients/families (23.5%) carried at least one of 232 these (mutations)’ Variants NOT ‘mutations’
- ‘The relationship between BRCA1 mutations and predisposition to neuroblastoma is based on casual findings in specific cases, such as our patient (10). Although the BRCA1-Neuroblastoma risk ratio is still under study, we consider this variant to be likely involved in neuroblastoma predisposition. The parents refused the family segregation study and no additional family information was provided. ‘ I seriously disagree with this ‘likely involved’ conclusion. This is a common founder variant carried by 1% of Jewish people. If the parents are Jewish this is a likely chance finding
- ‘ The NGS study identified heterozygous CHEK2 c.497A>G (p.N166S) NM_007194.3. Segregation study confirmed the maternal origin of the variant and the remaining members of the family are under study. The evidence available in breast cancer led us to suggest the involvement of the variant in the mother's breast cancer [40]. However, evidence supporting the relationship between CHEK2 variants and the risk of leukemia is still limited [41]. Even so, we considered this variant to be likely involved in susceptibility to leukemia in the pediatric age.’ This consultant in Medical Genetics again does not agree with this. Even the breast cancer link for missense variant is associated with only a 1.6-1.7 relative risk meaning most breast cancers will be coincidental. This is not good clinical governance to make these decisions that are likely misleading
- ‘On the basis of our results, it can be concluded that at least 9.4% patients have a genetic predisposition syndrome which explains the cancer they suffer’ A separate table of these 16 with references to the evidence base for them being definitively associated is required
- The abstract discussion and conclusions all need tempering to reduce the slight overenthusiasm. In particular the main highlight is actually the low useful detection rate of definitive genetic causality outside Jongmans. The lesson of this is that potentially more harm may be done by speculation about causality where there is little evidence especially outside Jongmans criteria cases.
Author Response
Thank you very much for the effort you put into reviewing our work. Your input has been very valuable.

Reviewer 2 Report
A very clearly structured an well-planned study on cancer predisposition syndromes in childhood. Clinical importance, methodology and results are presented in a very clear manner. However, regarding the discussions, I am missing some points / aspects.
The used Jongmans tool for the detection of CPS has been modified and adjusted by others. The LFS patient that was missed by the Jongmans tool would have been detected by the modified version by Ripperger et al (Am J Med Genet A., 2017; doi: 10.1002/ajmg.a.38142) which has been further developed by the GPOH group on CPS recently. So, this should be mentioned in the discussion.
The difficult topic of the predictive value of VOUS and VOPPS as well as incidental findings has to be adressed in the discussion as this is a very central issue when reporting VOPPS and thinking about screening for CPS without selection by a tool like the Jongmans or the Ripperger tool.
Author Response
Thank you very much for the review and for having positively valued our work

Round 2
Reviewer 1 Report
The authors present the results of germline NGS testing of an unselected series of children with malignant tumors including benign CNS tumors. The authors try to make to much out of probably chance findings of a common BRCA1 and CHEK2 variant as well as a number of other variants that are not definitively associated with the tumor being assessed.The main finding is actually that Jongmans’ criteria are extremely effective in identifying children who should receive a panel. With 30% detection rate of definitely associated genetic predisposition compared to only 1/120 of the remainder. Given the complexity of genetic counselling and potential for overinterpretation that the authors themselves have fallen into the main conclusion should be that testing outside Jongmans’ criteria should only be done on a research basis to determine potential links. The authors should provide more information on those with definite gene associations as to whether they met clinical criteria such as bilateral retinoblastoma, NF1 and NF2. The abstract discussion and conclusions all need tempering to reduce the slight overenthusiasm. Nevertheless, with a more careful assessment of the data and a change in emphasis this may still be a useful contribution. For instance the findings relating to the CTCF Variant c.1337-T > A and Wilms Predisposition is quite compelling and there are previous studies to suggest the potential importance of CCTF in Wilms. Overall the tenor of the entire paper and conclusions must be rewritten as is the main focus on stating how ‘useful’ the panels are.: Thank you very much for the comments. We have improved the paper based on this overview evaluation and on the specific comments provided.
The authors have not responded fully to the suggestions. The paper is still essentially unchanged apart from some toning down of the BRCA1 and CHEK2 classification and the conclusions. The main finding is that Jongman’s is useful and outside Jongman’s detection is VERY low
Specific comments
- ‘However, for several disorders, knowledge remains scarce which is the case of many soft tissue and bone sarcomas [15, 16].’ -It is well known that germline TP53 predisposes to both: Thank you very much. We agree with you about this comment and thereby, we have changed the position in the sentence for “soft tissue and bone sarcomas [15, 16]”.
OK
- ‘Genes involved in predisposition to the patient's tumor were studied as score 1. Genes involved in predisposition to other tumors, as score 2 and other genes in the panel were included in score 3’ Why were score 3 included in the panel if they were not known to be associated with childhood cancer?: It is an extremely reasonable question. The panel was initially designed for studying pediatric tumors and the germline of the patients in our hospital and therefore, not only for studying germline variants. This is the main reason for the inclusion of these genes in the design. Moreover, studying these genes from patient’s germline was considered an opportunity to investigate new genotype-phenotype relationships and thus, they were included in the technical analysis.
There is no alteration to the text and no real answer here
- ‘Finally, 170 patients fully met the inclusion criteria and agreed to participate in the study.’ -I presume it was the parents that agreed?: Yes, the legal guardian signed the informed consent in all cases, but the patients were also informed according to their ages and over 12 years signed specific documents.
You need to alter the text
- ‘Other mutated genes were DICER1, NF2, SUFU, TP53, XPC, SOS1, along with trisomy 21.’ Trisomy 21 is not a ‘mutated’ gene. Presumably this was known clinically? Was the panel actually used for this and did it pick this up?: We totally agree with this comment, it is not correct to include trisomy 21 next to “other mutated genes”. We have modified the sentence.
OK
- ‘Regarding the genes involved in predisposition, the most frequently altered was the RB1 gene (6/16; 37.5%) followed by NF1 (3/16; 18.8%)’ -How many were already diagnosed clinically eg bilateral RB and NF1.Very interesting question. The main clinical data of patients are included in supplementary data 2. We did not detail every clinical aspect because this was not the objective of the work but, the genetic diagnosis of the patients. Some patients were highly suspicious clinically or even fulfilled clinical criteria pre-test.
- Table S1 please state if retinoblastoma bilateral or multifocal. Please confirm if really a ‘schwannoma’ with NF1. Please confirm if an ‘isolated’ VS in NF2. Schwannomas are not a feature of NF1 these are nearly always neurifibroma-was this biopsied?:
-We have reviewed the clinical data of the retinoblastoma patients. These data have been included in table s1.
-Patient number 127 was diagnosed with CNS mass. This was the reason to be referred to pediatric oncology unit. The mass was biopsied twice. The results were: Biopsy - 30th May 2019: Right temporal mass, biopsy: Cellular schwannoma, WHO grade I.Biopsy - 3rd Nov 2020: Brain mass; cellular schwannoma, WHO grade I. The patient suffers other clinical data characteristics of NF1 syndrome.
Schwannoma diagnosis does NOT support an NF1 diagnosis. How did the patient meet NF1 criteria
-Patient number 150, was diagnosed with bilateral VS. We have included this aspect in table s1.
- ‘Pathogenic or likely pathogenic variants considered as no(t) involved in the tumor etiology by the Pediatric Cancer Predisposition Committee. 2B.’ -‘not’: accepted and modified
OK
- ‘Overall and considering all P/LP variants identified, related or not to genetic predisposition to patient´s tumor, 40 out of 170 patients/families (23.5%) carried at least one of 232 these (mutations)’ Variants NOT ‘mutations’: accepted and modified.
OK
- ‘The relationship between BRCA1 mutations and predisposition to neuroblastoma is based on casual findings in specific cases, such as our patient (10). Although the BRCA1-Neuroblastoma risk ratio is still under study, we consider this variant to be likely involved in neuroblastoma predisposition. The parents refused the family segregation study and no additional family information was provided. ‘ I seriously disagree with this ‘likely involved’ conclusion. This is a common founder variant carried by 1% of Jewish people. If the parents are Jewish this is a likely chance finding: Thank you for this comment. We agree with you, the sentence translated an optimistic conclusion. We have modified the sentence in order to keep a more conservative position and probably more realistic.
OK
- ‘ The NGS study identified heterozygous CHEK2 c.497A>G (p.N166S) NM_007194.3. Segregation study confirmed the maternal origin of the variant and the remaining members of the family are under study. The evidence available in breast cancer led us to suggest the involvement of the variant in the mother's breast cancer [40]. However, evidence supporting the relationship between CHEK2 variants and the risk of leukemia is still limited [41]. Even so, we considered this variant to be likely involved in susceptibility to leukemia in the pediatric age.’ This consultant in Medical Genetics again does not agree with this. Even the breast cancer link for missense variant is associated with only a 1.6-1.7 relative risk meaning most breast cancers will be coincidental. This is not good clinical governance to make these decisions that are likely misleading: we agree with this affirmation again. These sentences were not sustained in evidence and in fact, we have moved this information to the family very carefully, considering that the variant may not be contributing to their diseases.
OK
- ‘On the basis of our results, it can be concluded that at least 9.4% patients have a genetic predisposition syndrome which explains the cancer they suffer’ A separate table of these 16 with references to the evidence base for them being definitively associated is required: These 16 pathogenic variants or genomic alterations are included in table S1. They were categorized as pathogenic alterations and so, they are reported in this way in Table S1. They are pathogenic variants according to ACGM criteria. The technical and clinical analysis that established the functional classification is included in supplementary data 3. Honestly and respectfully, we disagree with your comments at this point. We think that the information available in the paper is enough and in fact, greater than that available in other similar papers.
I disagree
- The abstract discussion and conclusions all need tempering to reduce the slight overenthusiasm. In particular the main highlight is actually the low useful detection rate of definitive genetic causality outside Jongmans. The lesson of this is that potentially more harm may be done by speculation about causality where there is little evidence especially outside Jongmans criteria cases.: We agree with this opinion, especially the conclusions section had an optimistic tone that we have modified after the review.
Yet there is little change to the abstract or discussion and no mention of the low detection outside Jongman’s
Reviewer 2 Report
All my comments have been adressed.